# Fast Granulation by Combining External Sludge Conditioning with FeCl_3_ Addition and Reintroducing into an SBR

**DOI:** 10.3390/polym14173688

**Published:** 2022-09-05

**Authors:** Jun Liu, Shunchang Yin, Dong Xu, Sarah Piché-Choquette, Bin Ji, Xin Zhou, Jun Li

**Affiliations:** 1School of Modern Agriculture, Jiaxing Vocational & Technical College, Jiaxing 314036, China; 2Department of Civil Engineering, Tongji Zhejiang College, Jiaxing 314051, China; 3College of Geomatics & Municipal Engineering, Zhejiang University of Water Resources & Electric Power, Hangzhou 310018, China; 4Institute of Microbiology of the Czech Academy of Sciences, 14220 Prague, Czech Republic; 5Department of Water and Wastewater Engineering, School of Urban Construction, Wuhan University of Science and Technology, Wuhan 430070, China; 6College of Environmental Science and Engineering, Taiyuan University of Technology, Taiyuan 030024, China; 7College of Environment, Zhejiang University of Technology, Hangzhou 310014, China

**Keywords:** external conditioning, Fe^3+^ addition, reintroduction, extracellular polymeric substances, rapid granulation

## Abstract

The separation of light and heavy sludge, as well as the aggregation rate of floccular sludge, are two critical aspects of the rapid granulation process in sequencing batch reactors (SBRs) in the early stages. In this study, we investigated the impact of a method to improve both sludge separation and granulation by coupling effluent sludge external conditioning with FeCl_3_ addition and then reintroducing it into the SBR. By supplementation with 0.1 g Fe^3+^ (g dried sludge (DS))^−1^, the concentration of extracellular polymeric substances (EPS) and sludge retention efficiency greatly increased, whereas the moisture content and specific oxygen uptake rate (SOUR) sharply decreased within 24 h external conditioning. Aggregates (1.75 ± 0.05 g·L^−1^) were reintroduced into the bioreactor once daily from day 13 to day 15. Afterwards, on day 17, aerobic granules with a concentration of mixed liquor suspended solids (MLSS) of 5.636 g/L, a sludge volume index (SVI_30_) of 45.5 mL/g and an average size of 2.5 mm in diameter were obtained. These results suggest that the external conditioning step with both air-drying and the addition of Fe^3+^ enhanced the production of EPS in the effluent sludge and improved rapid aggregation and high sludge retention efficiency. Consequently, the reintroduced aggregates with good traits shortened the time required to obtain mature aerobic granular sludge (AGS) and properly separate light and heavy sludge. Indeed, this method jump-started the aggregation, and rapid granulation processes were successful in this work. Additionally, while the removal efficiency of chemical oxygen demand (COD) and nitrogen from ammonium (NH_4_^+^-N) decreased when reintroducing the treated sludge into the SBR, such properties increased again as the AGS matured in the SBR, up to removal efficiencies of 96% and 95%, respectively.

## 1. Introduction

In the last few decades, aerobic granular sludge (AGS) biotechnologies have been extensively studied and have shown great potential for wastewater treatment [1]. This was because of its numerous advantages of an excellent settleability, dense and sturdy microbial structures, lower operational costs and space requirements compared to activated sludge treatment [2,3,4]. One of the downsides of AGS is that its development from flocculent sludge to mature AGS can take several weeks or months, reducing its attractiveness as a biotechnological process and consequently, its more widespread use in wastewater treatment plants (WWTPs). In order to reduce the time required to obtain mature AGS, several parameters can be optimized, such as finetuning the hydrodynamic shear force applied or the inclusion of short feast–famine cycles [3]. Adjusting the operation conditions is therefore an effective strategy to improve the efficiency and maturation rate of AGS. There are two main types of improvements for AGS bioreactors: internal and external conditioning.

Internal conditioning methods have been widely used to reduce granulation time in recent years and consist of either selective pressure control or adding solid compounds as “nuclei” or carriers for rapid granulation. On the one hand, selective pressure control consists of decreasing the settling time, increasing organic loading rate (OLR) or shear force and prolonging the starvation period. A higher selective pressure can play a positive role in sludge granulation, but it can also negatively affect the long-term stability of AGS if applied during the initial stage [5,6]. On the other hand, supplying “nuclei” to act as inoculants can also prove greatly beneficial to AGS formation. These “nuclei” or inoculants can be diverse, namely including anaerobic granules [7], crushed granular sludge [8], aerobic granules in long-term storage [9], granular activated carbon (GAC) [10], dry sludge micropowder [11] or cultures of specific strains of *Klebsiella pneumoniae* and *Pseudomonas veronii* [12]. The main downside of these two methods is that they cannot be easily scaled up for use in WWTPs, but their adaptability is improving.

Meanwhile, several such “nuclei” have shown promising results at a lab-scale in synthetic wastewater. Namely, adding metallic ions (Ca^2+^, Mg^2+^, Mn^2+^) directly into wastewater accelerates AGS formation [13,14,15,16]. Studies have shown that iron salts (Fe^2+^/Fe^3+^) could play an important role in rapid AGS formation when mixed into the wastewater in a lab-scale sequencing batch reactor (SBR) [17,18]. Yu and collaborators [19] have reported that Fe^2+^ bind with extracellular polymeric substances (EPS) or S^2−^ in term of a tridimensional network or precipitate, served as nuclei accelerating microbial aggregation in wastewater treatment. However, the efficiency of microbial attachment and nitrogen removal decreased with higher Fe^2+^/Fe^3+^ concentrations [20]. Iron precipitates in excess also resulted in a decreased bioactivity of aerobic granules [17]. In other words, the aerobic granulation process takes more time if a higher concentration of iron ions is added during the start-up stage [21,22]. In this regard, dosage is key in order to optimize rapid AGS formation with iron ions. Consequently, the use of internal conditioning via the addition of iron ions directly into the influent has been always controversial. Furthermore, the addition of 0.5 mg·L^−1^ of magnetic powder (Fe_3_O_4_) with a sludge volume index (SVI_30_) of 28.5 mL·g^−1^ led to a shorter start-up time of aerobic granulation and to the formation of larger granules than with the same concentration of Fe^2+^ and Fe^3+^ when added into synthetic wastewater [18]. However, this more efficient addition comes at a cost, i.e., it will greatly increase the operating costs of WWTPs. More importantly, it makes the separation of light and heavy sludge more difficult due to the short settling time required, which can wash out aggregates and floccular sludge along with the effluent and is therefore counterproductive.

Therefore, the external conditioning of AGS has been proposed to circumvent these issues, namely, by effluent sludge collecting, dewatering and reintroducing the dewatered sludge into SBRs to quicken AGS formation [23,24]. This strategy not only greatly reduces the start-up time by providing a low airflow rate of 0.1 m^3^·h^−1^ (superficial gas velocity = 0.44 cm·s^−1^) but also preserves beneficial microbes (related to EPS production and organic degradation) from the effluent sludge enrichment and reuses them to enhance aerobic granulation [24]. However, the external conditioning of effluent sludge by natural drying [23] or by the addition of Ca^2+^ [24] leads to the formation of aggregates with a lower retention efficiency of 28% and 43%, respectively. Consequently, the feedback frequency of aggregates and the sludge granulation time are both prolonged. This implies that improving the retention efficiency of the reintroduced aggregates may have a significant impact on the formation of AGS.

Fe^3+^ is widely used in WWTPs as a coagulating agent due to its positive impact on flocculation and sedimentation in activated sludge [25,26]. Indeed, Fe^3+^ can bind itself to EPS and can promote the secretion of EPS in microbes [21,27,28]. To the best of our knowledge, both the addition of Fe^3+^ into the effluent sludge for external conditioning and the reintroduction of aggregates into the SBR for rapid granulation has yet to be tested in the early stages.

Therefore, the main objective of this study was to investigate the combined impact of the external conditioning of effluent sludge with FeCl_3_ addition, followed by its reintroduction into the SBR for rapid granulation in the start-up stage. Meanwhile, the physicochemical properties of AGS morphology, EPS content and size distribution were also measured and compared to external conditioning with the addition of FeCl_3_ and feeding back in this experiment.

## 2. Materials and Methods

### 2.1. Materials

The chemicals used on artificial wastewater (Table 1 and Table 2) and other reagents were all analytical reagents (AR), which were purchased from Sinopharm Chemical Reagent Co., Ltd., Shanghai, China in this experiment.

### 2.2. Reactor and Operating Conditions

A lab-scale SBR made of Plexiglas, with a dimension of 100 cm × 10 cm and a working volume of 4 L, was used in this study. A fine air pump (PL60, RESUN, Shenzhen, China) was used to supply aeration to the bottom of the SBR through air diffusers with an airflow rate of 0.1 m^3^·h^−1^ (superficial gas velocity = 0.44 cm·s^−1^) and a 50% volumetric exchange ratio. Each 160 min cycle consisted of the following steps: feeding for 5 min, aeration for 120 min, settling for 2–15 min, discharge for 5 min and idle time for 15–28 min. Reactor pumps and valves were controlled by a PLC (programmable logic controller) (Hangzhou Zhijiang Water Processing Equipment Co., Ltd., Hangzhou China) in the SBR. The temperature of the reactor was maintained at 20 ± 5 °C using industrial-grade air-conditioning (Gree Group Co., Ltd., Zhuhai, China), and the pH of the wastewater was maintained at 7.0–7.5 using 10% HCl. The fast start-up strategy used in this work combined the rapid AGS formation strategy [24] with an external conditioning step consisting of an addition of Fe^3+^ into the effluent sludge followed by its reintroduction into the SBR.

### 2.3. Seeding Sludge and Wastewater

The SBR was inoculated with 0.5 L of seeding sludge taken from a municipal sewage treatment plant (Qige WWTP, Hangzhou, China). The reactor was fed with an acetate-containing synthetic wastewater. The synthetic wastewater had an influent chemical oxygen demand (COD), nitrogen from ammonium (NH_4_^+^-N) and total phosphorus (TP) maintained at 500–600 mg·L^−1^, 25–32 mg/L and 10–15 mg·L^−1^, respectively. The wastewater manual, made once in three days, was stored in the sealed tank with a pump for stirring. A trace element solution was also added to the wastewater. The composition of the synthetic wastewater and its enrichment in trace elements can be found in Table 1 and Table 2, respectively.

### 2.4. FeCl_3_ Dosage Strategy

In order to evaluate the difference between adding a divalent (CaCl_2_) [24] or a trivalent metal ion (FeCl_3_) on the effluent sludge and the rapid AGS formation process, the same dosage of 0.1 g Fe^3+^·(g dried sludge (DS))^−1^ was used in this study. The aim was to evaluate the role of the same dosage of Fe^3+^ on the effluent sludge external conditioning process.

### 2.5. Analytical Methods

The COD, NH_4_^+^-N, nitrogen from nitrate (NO_3_-N), nitrogen from nitrite (NO_2_-N), mixed liquor (volatile) suspended solids (ML(V)SS), sludge volume index at 5 min and 30 min (SVI_5_ and SVI_30_) and specific oxygen uptake rate (SOUR) were measured according to standard methods [29]. Sludge morphology was regularly assessed using an optical microscope (Olympus CX31, Tokyo, Japan). Sludge size was determined with Image-Pro Plus software (Analysis 6.0, Olympus Soft Imaging Solution, Münster, Germany). Granular settling velocity was measured by recording the time necessary for granules to fall from a given height in a measuring cylinder. Moisture content (%) was calculated by subtracting dry sludge weight from the total sludge weight and then dividing this amount by the total sludge weight.

EPS was extracted from the sludge using the formaldehyde–NaOH method described in earlier work [30], and measurements were made using an ultraviolet spectrophotometer (TU-1810, Beijing, China). Protein (PN) concentration in EPS was determined using Coomassie brilliant blue using bovine serum albumin (BSA) as standard [31]. Polysaccharide (PS) concentration in EPS was determined using the phenol–sulfuric acid method [32] and using glucose as standard.

## 3. Results and Discussion

### 3.1. Effects of Directly Adding FeCl_3_ into the Effluent Sludge

Figure 1 shows the morphology of the effluent sludge from the SBR before and after the external conditioning with Fe^3+^ addition. On average, 0.610 ± 0.005 g·L^−1^ of loosely structured effluent sludge was washed out in the first 2 min of each cycle on day 10 (Figure 1a). The sludge was collected with a gauze sieve (0.1 mm mesh size) after 24 h. On day 11, the collected effluent sludge was supplemented with 0.1 g Fe^3+^·(g DS)^−1^. The floccular sludge began rapidly aggregating, resulting in compact red aggregates with a relatively uniform size during the air-drying process (Figure 1b), which then formed many aggregates under the action of being manually crushed in fresh water (Figure 1c).

The properties of the effluent sludge varied considerably during the external conditioning process (Figure 1d). EPS concentration increased from 109.4 mg·g^−1^ to 360.5 mg·g^−1^ VSS. PN and PS concentrations showed a similar trend as EPS; however, the concentration of PS was lower than PN at the beginning of the conditioning process, and it reached a higher concentration at the end. This is in agreement with previous work [28,29]. Contrastingly, the SOUR and moisture content decreased from 15.7 to 4.25 mg O_2_ g^−1^ VSS h and from 0.97 to 0.45 within 24 h, respectively. Such values are similar to those reported after performing an external conditioning step including air-drying and the addition of Ca^2+^ [23,24]. However, the EPS content in this study is twice as high as in previous works [23,24], which is presumably caused by a higher sensitivity of microbes to Fe^3+^ than to Ca^2+^ or to natural drying only [21,25]. Furthermore, Fe^3+^ had a stronger binding ability with EPS than Ca^2+^ [25,27]. Consequently, more compact and smaller aggregates formed (Figure 1c) due to interactions between Fe^3+^ and EPS and phosphorus-containing precipitates [28]. Moreover, the moisture content of the effluent sludge plummeted during the external conditioning process, which is a common phenomenon in sludge dewatering. Sludge began to aggregate shortly after the beginning of the conditioning process, which sharply decreased the SOUR. The now hypotonic environment stimulated the microbial production of EPS as a means of protection against osmotic stress [33], which consequently improved the rapid aggregation process. Overall, this external conditioning method led to a quicker sludge aggregation process than sludge conditioning via natural drying [23] or Ca^2+^ addition [24].

Another SBR (R2) was used in this study to determine the impact of hydraulic pressure on sludge aggregates after external conditioning (Figure 2). Sludge aggregates (Figure 2a) with a MLSS of 2.68 g/L and a mean size of 4 mm were used as seed culture, and freshwater was used as influent. The operating conditions of R2 were the same as in R1 on day 10.

The hydraulic pressure exerted by the reactor broke down larger aggregates into aggregates and floccular sludge. The resulting aggregates were retained in the reactor, while the flocs were washed out after a 2 min settling time (Figure 2b). The majority of aggregates kept a compact structure throughout the long-term operation of R2, although a few of them were still loose or filamentous on day 7 (Figure 2c). The latter could be caused by a low food/micro-organism (F/M) ratio, which promotes a filamentous growth in aggregates [1,3]. Indeed, MLSS decreased from 2.68 g·L^−1^ to 1.9 g·L^−1^ while the sludge-retaining efficiency reached up to 0.71 on day 7 (Figure 2d). The size distribution of sludge aggregates was also investigated on days 0, 3 and 7 (Figure 2e). The average size of aggregates in the effluent sludge (Figure 1a) increased to 1.0–1.5 mm in the first 24 h following the external conditioning, while it decreased to 0.5–1.0 mm in the following days (Figure 2e). Indeed, the aggregates quickly absorbed water after their reintroduction into the SBR, which also decreased the overall sturdiness of the sludge. Therefore, the hydraulic shear force broke down more fragile aggregates into smaller aggregates and flocs. Consequently, the proportion of aggregates with a size between 1.0–4.0 mm decreased, while the proportion of aggregates smaller than 1.0 mm drastically increased on day 3 and by even more on day 7 (Figure 2e). Finally, most aggregates with compact structure remained in the SBR, while a few flocs were washed out following a 2 min settling time in the reactor. This indicates that aggregates with a higher EPS content, which are sturdier, can also withstand hydraulic pressure more than what was reported in earlier works [23,24].

### 3.2. Rapid Formation of AGS

Throughout the rapid granulation process, the most easily observable changes in sludge morphology were related to their color and shape. Initially, the seeding sludge was black, while its darker hue gradually faded within the first 7 days and then turned to a yellow color from day 7 to day 10. Additionally, during the first week process, sludge had a loose, irregular shape that was undistinguishable from floccular sludge (Figure 3a). On day 12, a few aggregates were observed, but the flocs were still dominant in part due to the decrease in settling time from 15 to 2 min (Figure 3b). On day 13, 1.75 ± 0.05 g/L aggregates of the external conditioned with Fe^3+^ were fed back into the SBR, which led to the formation of denser and more clearly defined aerobic granules in the following week (Figure 3c). Simultaneously, larger aggregates were broken down into more numerous, albeit smaller aggregates, which acted as nuclei or carriers for accelerating sludge granulation (Figure 3d,e). On day 57, aerobic granules with a compact structure and a yellow color could be seen with the naked eye (Figure 3f). In summary, the external conditioning using Fe^3+^ followed by the reintroduction of sludge aggregates into the SBR contributed to a quicker and more efficient rapid granulation process compared to internal conditioning with increasing selection pressure [5], improving wastewater temperature [34] or directly adding metal ions into artificial wastewater [13,14,15,16].

The MLSS and MLVSS/MLSS ratio in the SBR showed an increasing trend (3.75–5.88 g·L^−1^ and 0.5–0.82) in the first 10 days (Figure 4a). MLSS and MLVSS/MLSS sharply decreased on day 10 following the reduction in settling time and increase in the amount of sludge washed out from the SBR. Then, the conditioned aggregates (1.75 ± 0.05 g·L^−1^) were gradually fed back once into the SBR from day 13 to day 15. This led to a quick increase in MLSS and MVLSS, followed by their stabilization at 6.28 g·L^−1^ and 5.72 g·L^−1^ (Figure 4a), respectively. This could be caused by the rapid formation of sludge aggregates from the external condition with the addition of FeCl_3_, which, in turn, slightly increased the sludge ash content. This suggests that conditioning the effluent sludge by adding 0.1 g Fe^3+^·(g DS)^−1^ followed by its reintroduction into the reactor accelerated the granulation process without significantly altering the ash content and the bioactivity of AGS. This is in line with the SOUR profile shown in Figure 4d, which depicts an increase in SOUR from day 0 to day 12, followed by a sharp decrease when conditioned sludge was reintroduced into the SBR between day 13 and 15 and then by a gradual increase and stabilization of SOUR at 24 mg O_2_·(g VSS·h)^−1^ from day 16 onwards. Compared to earlier works [17,35], another advantage of this conditioning method is that it does not lead to the precipitation of excess metallic ions during the start-up stage in this experiment, which consequently maintains the ash content at a lower level throughout the granulation process. (MLVSS/MLSS value had not greatly changed).

SVI is an important index used to measure the settling properties of the sludge. Five days post inoculation of the seeding sludge, both SVI_5_ and SVI_30_ values greatly increased (Figure 4b). SVI_5_ and SVI_30_ values plummeted from 159.7 and 77.3 mL·g^−1^ on day 10 to 43.1 and 43.1 mL·g^−1^ on day 15, respectively, coinciding with the reintroduction of the conditioned sludge. Both SVI indices stabilized at 45.0 mL·g^−1^ following the development of aerobic granules. Correspondingly, the SVI_30_/SVI_5_ ratio also increased from 0.48 on day 10 to 1.0 on day 15, which hints at the successful formation of AGS [24,36]. Such results show that it is possible to achieve AGS by external conditioning with FeCl_3_ addition and reintroducing compact aggregates into the SBR, making the rapid granulation process shorter by 3–7 days than in previous works [25,26] during the start-up stage. Indeed, the addition of Fe^3+^ as external conditioning increased the retention efficiency to 0.71 (Figure 2d) compared to natural drying (0.28) and the addition of Ca^2+^ (0.43) [23,24]. This likely had a direct impact on the aerobic sludge granulation. Additionally, settling velocity can also reflect biomass settleability. As seen in Figure 4d, settling velocity slightly decreased in the first 10 days and quickly increased after the reintroduction of compact aggregates starting on day 13. With the successful development of AGS, the settling velocity stagnated from day 23 onwards at a value of approximately 30 m·h^−1^.

EPS, produced by microbes, are crucial components of the aerobic granulation process. The concentration of EPS as a function of time in the SBR is shown in Figure 4c. The EPS concentration slightly increased in the first 10 days and then decreased between day 10 and 12 due to the reduction in settling time from 15 to 2 min. After the aggregates were reintroduced into the SBR from day 13 to 15, the EPS content sharply increased to 153.0 mg·(g VSS)^−1^ on day 15. This rise in EPS is likely due to the reintroduction of conditioned EPS-rich sludge aggregates into the SBR. Afterwards, EPS slightly decreased to 132 mg (g VSS)^−1^ from day 17 onwards. The concentration of PN and PS showed a similar trend. At the end of the SBR operations on day 57, EPS, PN and PS concentrations were 128.2 mg·(g VSS)^−1^, 65 mg·(g VSS)^−1^ and 63.2 mg·(g VSS)^−1^ (Figure 4c), respectively. These results suggest that the reintroduction of conditioned sludge with a high EPS content can further increase EPS production, which, in turn, accelerates AGS formation in the SBR, as observed in earlier works [23,24].

### 3.3. SBR Treatment Performance

The COD and NH_4_^+^-N removal efficiencies were also investigated throughout the AGS maturation process. Figure 5a shows that influent COD and NH_4_^+^-N start at a concentration of 500–600 mg·L^−1^ and 25–30 mg·L^−1^, respectively. However, the removal rate of each compound of interest widely varies throughout the granulation process, which is why the SBR seems less than optimal if data from day 0 to day 57 are taken into account (Figure 5a). Indeed, biomass concentration rapidly increases in the first 10 days of operation so that removal rates increase, while the reduction in settling time on day 10 temporarily disrupts the efficiency of the reactor. Consequently, the COD and NH_4_^+^-N skyrocketed from 45 mg·L^−1^ and 3.61 mg·L^−1^ on day 10 to over 104 mg·L^−1^ and 6.6 mg·L^−1^ on day 12 and then peaked at 184 mg·L^−1^ and 12.3 mg·L^−1^ on day 14. The sludge conditioned with Fe^3+^ and reintroduced into the SBR had a low SOUR (Figure 1d), which could have decreased the removal rates of both COD and NH_4_^+^-N. The SOUR parameter increased a few days later and then stabilized during the rapid granulation process, which resulted in the elimination of the remaining COD and NH_4_^+^-N by the end of each cycle and therefore, a removal efficiency of 100%.

Figure 5b shows that the variations of COD, NH_4_^+^-N, NO_3_^−^-N and NO_2_^−^-N in one type of cycle after the AGS reached maturity on day 40. In fact, the results show that COD is almost entirely eliminated after 30 min of operation (Figure 5b), presumably due to the high amount of MLSS (Figure 4a). On the other hand, the NH_4_^+^-N is completely eliminated after a longer time, i.e., 90 min of operation. An accumulation of NO_3_^−^-N is also observed after 60 min, yet its concentration eventually decreases to 0 mg/L after 120 min. The NO_2_^−^-N has a smaller variation in concentration but follows a similar trend as NO_3_^−^-N (Figure 5b). These results depict that simultaneous nitrification and denitrification (SND) occur in the SBR. In summary, a complete removal of COD and NH_4_^+^-N was achieved more quickly in this study than reported in previous work using the external conditioning method [25,26]. The addition of Fe^3+^ within the external conditioning process not only reduces sludge granulation time but also decreases the time required to completely eliminate NH_4_^+^-N and COD by the end of each 120 min cycle by favoring the growth of slow growing organisms (e.g., nitrifiers and denitrifiers) in the bioreactor.

### 3.4. Hypotheses Regarding the Improved Rapid Granulation Process

Table 3 shows several properties of the granular sludge and its granulation process according to the external conditioning method used, based on this study and two previous ones. Few studies using external conditioning and reintroduction steps have been published so far, indicating that much work still has to be undertaken in the future. To the best of our knowledge, only the air-drying [23], CaCl_2_ addition [24] and FeCl_3_ addition (this work) methods have been tested as effluent sludge external conditioning methods, all of which were performed over 24–72 h. While the difference in several sludge properties between methods was negligible, the concentration of EPS and the retention efficiency showed more pronounced differences. Indeed, EPS increased as sludge retention efficiency increased, as shown in Figure 6a, depicting a positive relationship between the two variables (retention efficiency = 0.0019 × EPS + 0.0288, R^2^ = 0.9375). EPS likely play a key role in the aggregation of effluent sludge during the external conditioning process. In addition, granulation time decreased as the retention efficiency increased (Figure 6b), i.e., showing a negative trend (where granulation time = −13.541 × retention efficiency +26.409, R^2^ = 0.9704). In summary, the addition of Fe^3+^ was the most efficient conditioning method out of all three, with an up to twofold increase in EPS concentration and retention efficiency. Indeed, studies have shown that Fe^3+^ has a higher tendency to form ionic cross-linked hydrogels with alginate compared to Ca^2+^ [22,27,37]. Furthermore, micro-organisms could be more sensitive to Fe^3+^, thus increasing their production of EPS following the conditioning process [20,22,27,38].

A proposed model of the external conditioning used in this study can be found in Figure 7. Briefly, following the natural drying of floccular sludge, the addition of FeCl_3_ and the reintroduction of the sludge into the SBR, the EPS concentration quickly increased (Figure 1d). The latter could, in turn, enhance microbe–microbe or microbe–sludge adhesion, which then further promotes cohesiveness within the sludge [33,38]. The sludge dewatering and drying process might also shorten the distance between microbes in the flocculent sludge, which could improve the microbial aggregation process. Additionally, some microbes could be more sensitive to the presence of Fe^3+^ than others, causing them to secrete more EPS to protect themselves against a harsher environment [20,33,37,38]. Consequently, the floccular sludge (effluent sludge) formed into aggregates. Then they were crushed into aggregates with compact or loose structure with the action of hydrological shear force and afterwards, returned back into the SBR. The compact aggregates with a high concentration of EPS and retention efficiency (Figure 1 and Figure 2) stayed in the SBR to act as nuclei or carriers for microbial latching on and proliferation. Finally, AGS was rapidly formed. Meanwhile, the loose aggregates were broken up into the flocs by the shear force, discharged out of the SBR with 2 min setting time. And the floccular sludge was collected for external conditioning with FeCl_3_ addition then fed back into the bioreactor for improving rapid granulation. This process was until complete aerobic granulation (Figure 7). Briefly, this work proved that the reintroduction of aggregates was facilitated by an easy separation of floccular sludge (light sludge) and aggregates (heavy sludge) in the SBR with a 2 min settling time. The concentration and enrichment of EPS-producing microbes in the reintroduced sludge further quickened the rapid granulation process. Consequently, improving rapid granulation with this method was successful in this experiment.

## 4. Conclusions

In this study, the fast formation of aerobic granules was achieved on day 17 using effluent sludge external conditioning with the addition of Fe^3+^ and reintroduction into the SBR. Adding Fe^3+^ to the effluent sludge can quickly improve EPS content and high retention efficiency and decrease SOUR and moisture content within a 24 h external conditioning process. This method greatly improved the rapid aggregation compared to natural drying or Ca^2+^ addition. The reintroduced aggregates quicken the AGS formation through acting as nuclei or carriers in the SBR with a 2 min settling time. In short, the strategy used in this work was proved to be feasible for enhancing fast granulation.

## Figures and Tables

**Figure 1 polymers-14-03688-f001:**
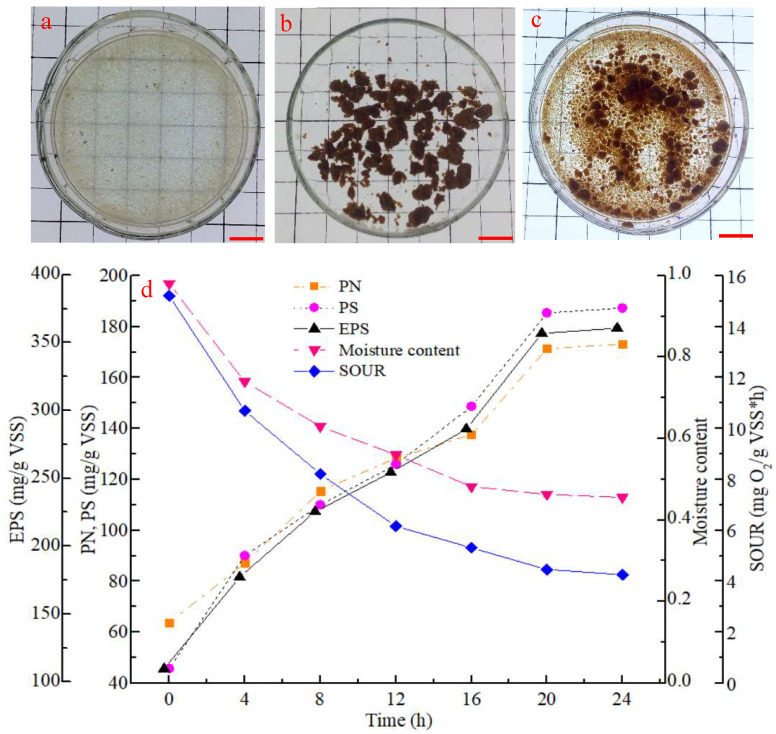
Evolution of sludge physicochemical properties throughout the external conditioning process with FeCl_3_: (**a**) raw effluent sludge, (**b**) sludge after 24 h of external conditioning with FeCl_3_ and air-drying, (**c**) sludge manually crushed in fresh water, and (**d**) variations in PN, PS, EPS, SOUR and moisture content throughout the conditioning process. Scale: 1 cm.

**Figure 2 polymers-14-03688-f002:**
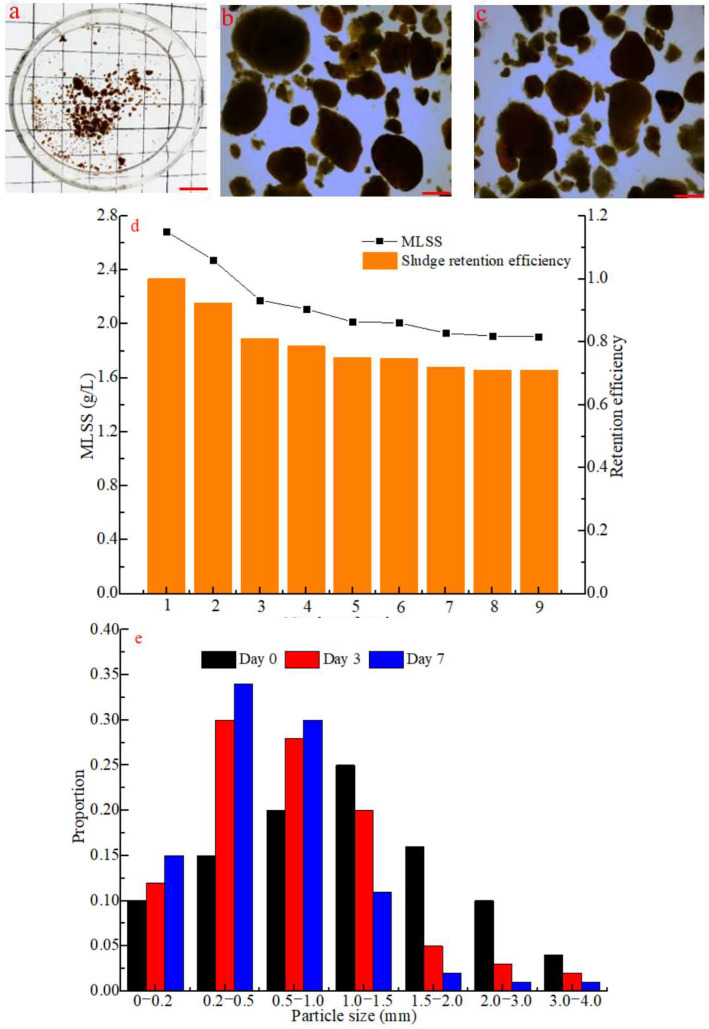
Effect of hydrodynamic force on sludge aggregates on (**a**) day 0, (**b**) day 3 and (**c**) day 7. (**d**) Sludge retention efficiency and concentration of suspended solids in the SBR as a function of the number of completed cycles. (**e**) Size distribution of aggregates within the first week post-conditioning; scale: 1 cm.

**Figure 3 polymers-14-03688-f003:**
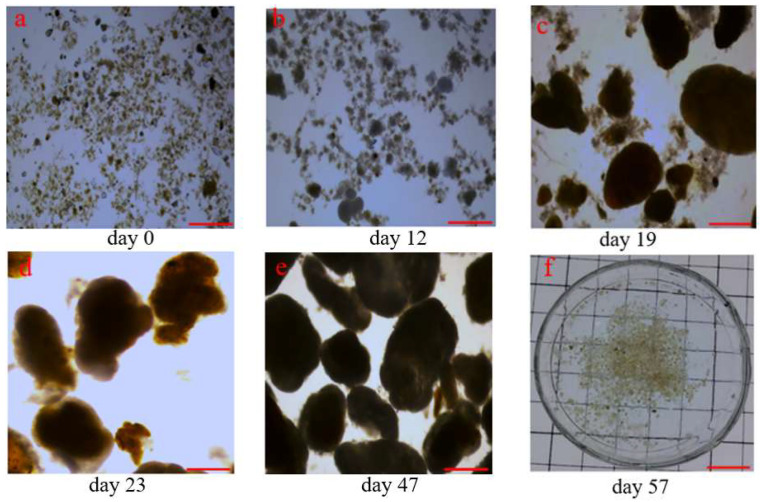
Photographs of the granular sludge during the rapid granulation process; scale bars = 1 cm.

**Figure 4 polymers-14-03688-f004:**
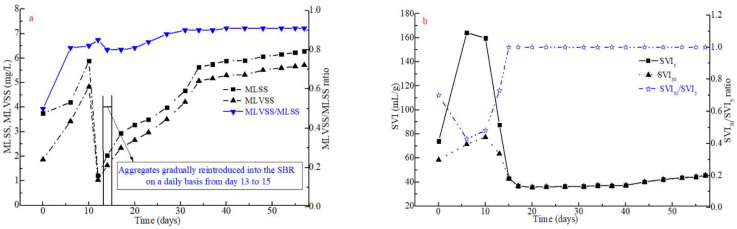
Evolution of sludge particles throughout the rapid granulation process in regard to their (**a**) MLSS and MLVSS, (**b**) SVI_5_ and SVI_30_, (**c**) EPS, (**d**) SOUR and settling velocity and (**e**) size distribution.

**Figure 5 polymers-14-03688-f005:**
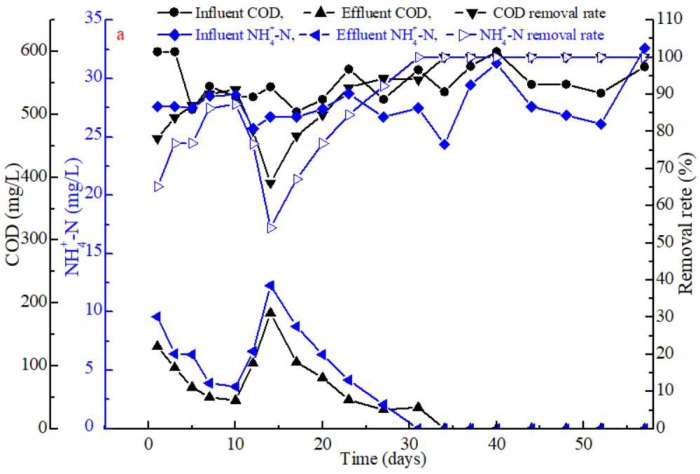
Reactor performance profile during (**a**) the entire rapid granulation process and (**b**) during a single day after the stabilization of the granulation process on day 40.

**Figure 6 polymers-14-03688-f006:**
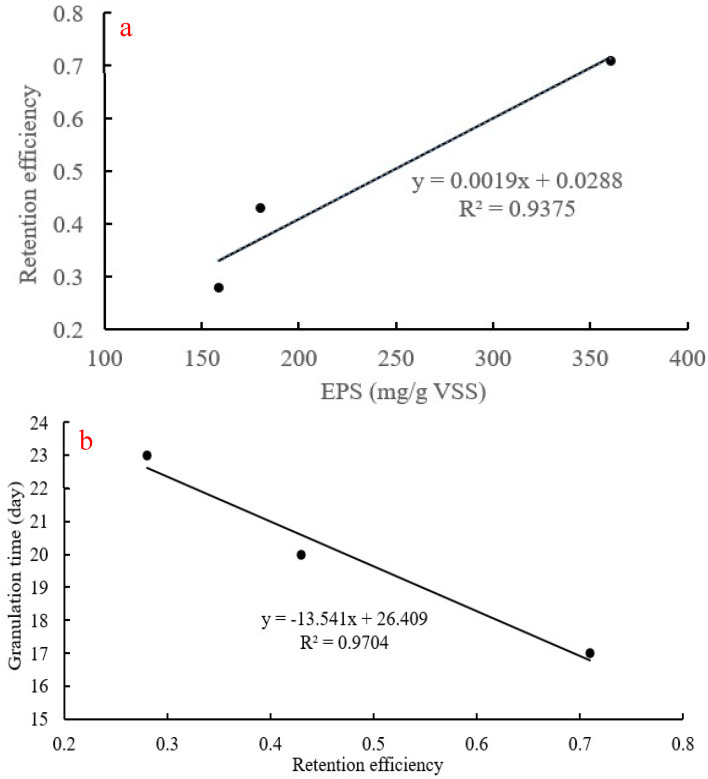
Relationship between (**a**) aggregates retention efficiency and (**b**) between retention efficiency and granulation time.

**Figure 7 polymers-14-03688-f007:**
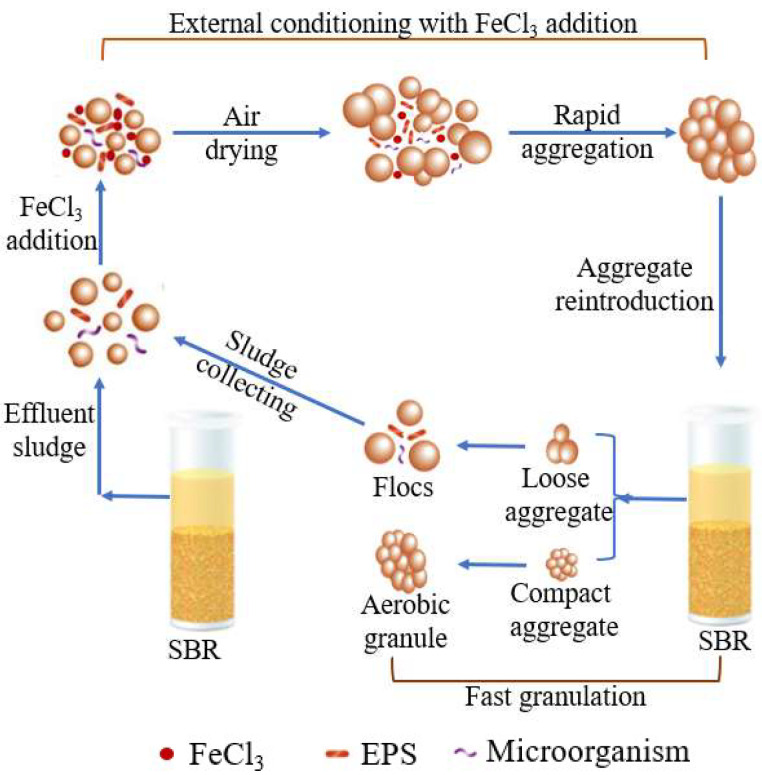
Proposed rapid granulation model using external conditioning with the FeCl_3_ addition and the reintroduction of aggregates in an SBR.

**Table 1 polymers-14-03688-t001:** Detailed composition of the synthetic wastewater.

Composition	Concentration (mg·L^−1^)	Composition	Concentration (mg·L^−1^)
NaAc	1200	CaCl_2_	21
NH_4_Cl	115	MgSO_4_·7H_2_O	102
KH_2_PO_4_	63	Trace element	0.5 mL·L^−1^
FeSO_4_·7H_2_O	8		

**Table 2 polymers-14-03688-t002:** Concentration of trace elements in the SBR.

Composition	Concentration (g·L^−1^)	Composition	Concentration (g·L^−1^)
H_3_BO_3_	0.2	Na_2_Mo_7_O_24_·2H_2_O	0.14
CoCl_2_·6H_2_O	0.2	ZnSO_4_·7H_2_O	0.2
CuSO_4_·5H_2_O	0.06	KI	0.06
FeCl_3_·6H_2_O	2	NiCl_2_	0.12
MnCl_2_·2H_2_O	0.22		

**Table 3 polymers-14-03688-t003:** Studies testing the efficiency of the rapid granulation process using effluent sludge external conditioning followed by its reintroduction into SBR.

External Conditioning Method	Sludge Properties during External Conditioning Process					Rapid Granulation		Study
	Time (h)	SOUR (mg O_2_ ·(VSS·h)^−1^)	MC	EPS (mg·(g·VSS)^−1^)	Retention Efficiency	Reintroduction Frequency	Time (d)	
Natural drying	72	5.26	0.52	**158.6**	**0.28**	7	**23**	[23]
Ca^2+^ addition	24	4.3	0.55	**180.3**	**0.43**	5	**20**	[24]
Fe^3+^ addition	24	4.25	0.45	**360.5**	**0.71**	3	**17**	This study

MC-moisture content.

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
