# Peer review of "Fast Granulation by Combining External Sludge Conditioning with FeCl3 Addition and Reintroducing into an SBR"

_polymers, 2022, doi:10.3390/polym14173688_

Round 1
Reviewer 1 Report
Manuscript entitled “Improvement of the rapid granulation rate and sludge retention efficiency combining external sludge conditioning and dewatered sludge reintroduction into an SBR” submitted by Jun Liu, Shunchang Yin, Dong Xu, Sarah Piché-Choquette, Bin Ji, Xin Zhou and Jun Li, can be considered for publication in Polymers Journal, after a major revision.
Here is a list of my specific comments:
1. General comment 1: The utility of this study should be clearly highlighted in the manuscript.
2. General comment 2: All abbreviations should be explained.
3. Page 1, Abstract: Include in this section the most important results to highlight the importance of this study.
4. Page 2, 1. Introduction: In this section, the most important aspects related to this topic should be clearly presented to provide a properly description of the state of art in this field.
5. Page 2, line 51: “treatment [1](Adav et 51 al., 2008).” Please correct this.
6. Page 3, line 116: “Therefore, this study…”. At the end of Introduction, the main objectives of this study should be clearly and detailed presented.
7. Page 4, 3. Results and discussion: This section should be reorganized. The experimental should be clearly and detailed discussed in accordance with the main objectives of this study.
8. Page 15, 4. Conclusions: This section is too general. Included in this section the most important experimental results and findings to highlight the importance of this study.
Author Response
Dear reviewer, we carefully answered all questions by point to point and earnestly revised the manuscript based your suggestions, you can refer it from the attachment. Thanks and best wishes.

Reviewer 2 Report
Title: Improvement of the rapid granulation rate and sludge retention efficiency by combining external sludge conditioning and de-watered sludge reintroduction into an SBR
##Overall comments
The Paper describes the coagulation enhancement using FeCl3 in wastewater treatment at a specific time during the process. The Paper is interesting. However, this type of work has already been published by authors. Therefore, the novelty of this work is low. The Paper contains grammatical mistakes, and the presentation of figures is not sound. The weakness of the Paper is that there is no study on the biodegradation of the end product. The Paper needs a Major revision.
Major comments:
##Comments on title, abstract, and reference
1. The title needs revision.
2. The abstract should present the background, drawbacks, objectives, results, and utility. Authors should avoid writing repeating sentences and abbreviations in the abstract.
3. References should be in journal format. Among 41 references, the authors have cited six references of their work.
4. Keywords should be in words, not in abbreviations or formulas.
##Comments on Introduction
5. Line 51: Please delete (Adav et 51 al., 2008), Line 52-55: hefty sentences and check English. Line 87-88: "the aerobic granulation process takes more time if a higher concentration of iron ions is added during the startup stage" Why?
6. Line 106: Why is Fe (III) better than Ca (II)? Is there any report about using Al (III)? Please include the clarification.
7. What is the novelty of this study compared to References number 25: https://doi.org/10.1016/j.chemosphere.2019.125159
8. The authors should mention the novelty of this study compared to their previous work.
9. The study's objective should be more apparent to the reader.
##Comments on Materials and method
10. The details of materials used in this study may be spared in a section named materials. What was the pH of wastewater before adjustment? What is the name of salts added for Fe (III)?
11. Line 133: How did the authors measure the trace elements in wastewater?
12. Line 141: How did the authors maintain the COD and nitrogen in wastewater?
13. Line 143: Did the authors use synthetic or real wastewater for this study?
14. Line: 150: 0.1 g metal iron/g...What is it?
##Comments on results and discussion
15. Line:205: Is it 'broke-down' or reduced in size from larger to smaller minor?
16. Fig.1d should be redrawn using color line + symbols
17. Fig. 4 should be merged into a page.
18. The numbering of Fig. into the text should be in chronological order, e.g., line 260 Figure 40a and then line 264 Fig. 5a. It would be: Fig. 4b, 4c, 4d, de and so on. after that Figure 5. Please check the entire manuscript.
19. Line 223:The addition of Fe3+ might have stimulated microbes...Why?
How did the authors analyze the surface morphology of sludge (e.g., Fig. 1a-1c)?
20. Fig. 7 represents a chain process? is it proper for this study?
21. What will biodegradation of Fe (III) mixed sludge be? Is there any study? What types of microbes show more growth in the presence of iron? Is there any report?
22. What percentage of iron is in the final sludge? If possible, the reviewer suggests SEM and EDS analysis of the end product.
##Comments on conclusion
23. The conclusion has been written per the study's findings. However, "aggregates could then act as nuclei promoting the attachment and growth of microbes on their surface, which quickened the AGS formation process" is not clear from this study.
Author Response
Dear reviewer, we carefully answered all questions and seriously revised the manuscript, you can refer it from the attachment. Thanks and best wishes.

Round 2
Reviewer 1 Report
All my previous remarks and comments have been considered into new version of the manuscript. It means that reviewed manuscript meets the criteria and in my opinion can be published as original paper in Polymers Journal.
Author Response
Dear sir,
The manuscript entitled “Fast granulation by combining external sludge conditioning with FeCl3 addition and reintroducing into an SBR” is revised based on your suggestions,you can refer it from the attachment on "the revised manuscript"
Thanks.

Reviewer 2 Report
Dear Authors,
Thank you for responding to my questions, but I only found 50-100 word changes in the manuscript. The reviewer compared your revised manuscript to your previous one. Please do not believe that reviewers will be blind for the second time. You have used a yellow highlighter maximum for no changes at all. The Figures are not well organized. Because of your careless review, I recommend that your manuscript not be accepted and published in this prestigious journal.
Best wishes!
Author Response
Dear sir,
The manuscript entitled “Fast granulation by combining external sludge conditioning with FeCl3 addition and reintroducing into an SBR” is revised based on your suggestions, such as the manuscript title, abstract, introduction, results and discussion, conclusions and so on. The figures were checked, especially, figure 7 on the proposed model redrew again in the manuscript. The revised parts highlighted with blue (the first revision) and yellow (the second revision) in manuscript.
Thanks very much

Round 3
Reviewer 2 Report
Thanks for the corrections.
Best wishes!